# Factors Influencing Childhood Influenza Vaccination: A Systematic Review

**DOI:** 10.3390/vaccines12030233

**Published:** 2024-02-23

**Authors:** Kaiyi Han, Zhiyuan Hou, Shiyi Tu, Mengyun Liu, Tracey Chantler, Heidi Larson

**Affiliations:** 1School of Public Health, Fudan University, Shanghai 200437, China; kaiyi.han@lshtm.ac.uk (K.H.); sytu@fudan.edu.cn (S.T.); 2Department of Infectious Disease Epidemiology, London School of Hygiene & Tropical Medicine, London WC1E 7HT, UK; mengyun.liu@ucl.ac.uk (M.L.); heidi.larson@lshtm.ac.uk (H.L.); 3NHC Key Laboratory of Health Technology Assessment, Fudan University, Shanghai 200437, China; 4Department of Global Health and Development, London School of Hygiene & Tropical Medicine, London WC1E 7HT, UK; tracey.chantler@lshtm.ac.uk; 5Department of Health Metrics Sciences, University of Washington, Seattle, WA 98195, USA

**Keywords:** child, influenza, vaccine

## Abstract

Childhood influenza vaccination coverage remains low in lower/middle-income countries. This systematic review aims to identify influencing factors around childhood influenza vaccination. A systematic literature review was conducted and included empirical studies with original data that investigated factors influencing childhood influenza vaccination. We searched MEDLINE, Web of Science, EMBASE, CINAHL Plus, Global Health, PsycINFO, and two Chinese databases, China Knowledge Resource Integrated Database and Chongqing VIP, using a combination of the key terms ‘childhood’, ‘influenza’, ‘vaccination’, and related syntax for all peer-reviewed publications published before December 2019. Thirty studies were included in the analysis. Childhood influenza vaccination was positively associated with caregivers’ knowledge of influenza vaccine, positive vaccine attitudes, self-efficacy, perceived susceptibility and severity of influenza, believing in the efficacy of influenza vaccine, the worry of getting sick, healthcare workers’ recommendations, and previous influenza vaccination experiences. Barriers included the fear of safety and side effects of the vaccine, as well as poor access to vaccination service. To improve childhood influenza vaccine uptake, health education is necessary to address caregivers’ lack of confidence on vaccine safety. Future studies are needed to investigate influencing factors around healthcare workers’ vaccination recommendation behaviors and the impact of contextual factors on public vaccination behaviors.

## 1. Introduction

Influenza is a highly infectious respiratory illness characterized by acute onset of symptoms including fever and cough that can result in serious complications (e.g., pneumonia, dehydration, and encephalopathy) and even death [1]. It causes considerable disease burden in terms of excessive morbidity, mortality, and hospitalization [2]. According to the global annual influenza-associated respiratory deaths register, 290,000–650,000 seasonal influenza-associated respiratory deaths (4.0–8.8 per 100,000 individuals) occurred annually from 1999 to 2015 [3]. One of the particularly vulnerable groups is young children, as they have the highest rates of infection in the community [4]. Influenza viruses cause great disease burden among children below 5 years of age, with an estimated 870,000 hospitalizations and 10,200 deaths per year worldwide [5]. To reduce transmission across all age groups and decrease population-level disease burden, the World Health Organization (WHO) recommends that children aged between 6 and 59 months are vaccinated against influenza annually [6]. Over 40% of countries list seasonal influenza vaccination in their National Immunization Schedule, including most countries across North and South America, Europe, and some countries in African, South-East Asia, and the West Pacific Region [7,8,9,10,11,12,13]. Meanwhile, influenza vaccination is not included in the National Immunization Program (NIP) in many countries. For example, in China, seasonal influenza vaccination must be purchased by recipients [14], and the vaccine uptake for the entire population is 1.9% [15]. Data collected between 2009 and 2012 indicate that influenza vaccination uptake among children <5 years of age living in five provinces in mainland China was about 26.4%, which is far from satisfactory [16].

Numerous factors contribute to the low influenza vaccination coverage in countries and regions. The WHO’s Strategic Advisory Group of Immunization (SAGE) proposed in the Vaccine Hesitancy Determinants Matrix that individual/social influences, contextual influences, and vaccine and vaccination-specific issues all play a role [17]. In addition to parents’ perceptions around vaccines, communication and information sources, access issues, cost, or travel time can all influence vaccination decision-making. Hence, these factors should be considered when investigating childhood influenza vaccination. Previous studies have primarily focused on individual-level factors, including knowledge of influenza, awareness of risk, and misconceptions regarding vaccine safety and efficacy.

As a result, this systematic review aims to summarize the available evidence in order to identify influencing factors around childhood influenza vaccination. The findings can inform further studies on factors influencing childhood influenza vaccination in lower/middle-income countries, like China, and finally contribute to country-level policy decisions and improve childhood influenza vaccination uptake.

## 2. Materials and Methods

### 2.1. Data Sources and Searches

We followed the Preferred Reporting Items for Systematic Reviews and Meta-Analyses statement guidelines for reporting systematic reviews in structuring the review findings [18]. This review aimed to identify determinants for childhood influenza vaccination globally. We systematically searched the following databases: MEDLINE, Web of Science, EMBASE, CINAHL Plus, Global Health, and PsycINFO; furthermore, in order to include more local studies in China, we searched two Chinese databases (China Knowledge Resource Integrated Database and Chongqing VIP). The search strategy was a combination of the key terms ‘childhood’, ‘influenza’, ‘vaccination’, and related syntax for all peer-reviewed publications published before November 2019; studies related to childhood influenza vaccination published later were pushed through those databases and included in the review for analysis after being confirmed to meet the criteria.

As a primary outcome of interest, “childhood influenza vaccination” indicated children’s flu vaccination status in the latest flu season. Relevant outcomes also included caregivers’ intention to vaccinate their children in the upcoming flu season.

### 2.2. Study Selection

Appendix A presents the search strategy. Quantitative data from cross-sectional and longitudinal studies, where relevant confounders were accounted for by the study design or analysis, were included. Qualitative studies were included in this review if the methods of data collection and analysis were eligible for inclusion. Experiments that generated empirical data were included whereas non-empirical studies or studies not reporting original data were excluded. We also excluded studies that focused only on knowledge, attitudes, and beliefs regarding childhood influenza vaccination, but did not refer to actual vaccine uptake. Two independent researchers (HKY and LMY) first screened titles and abstracts, and then scrutinized the full texts to estimate their eligibility. Any discrepancies in the process were resolved through discussion with a third reviewer until consensus was reached (see Figure 1).

### 2.3. Data Extraction and Quality Assessment

A standardized form based on Cochrane Review and behavioral theories including the ‘Knowledge, Attitude, and Practices’ model, the Health Belief Model, and the Theory of Planned Behavior was developed specifically for this review prior to data extraction. Topics in the form include knowledge and awareness of influenza and influenza vaccines, confidence in the importance, safety, and efficacy of influenza vaccines, perceived susceptibility and severity of influenza, benefits and barriers to influenza vaccination, cue to action and social norms, self-efficacy, and emotions. Data were double extracted by two reviewers (H.K.Y. and L.M.Y.). The information extracted included characteristics of the study, methods, target population, sample size, childhood influenza vaccination, and associated factors influencing behaviors. Numerical data (numbers or percentages) that reported prevalence and non-medical factors of influenza vaccination were extracted from the quantitative component; themes relevant to factors influencing vaccination behaviors were extracted for the qualitative component. Factors that were examined as the predictors of influenza vaccination uptake among respondents will be presented in the Results section.

Two reviewers (H.K.Y. and L.M.Y.) independently assessed the quality of the included studies. The adapted BMJ survey appraisal tools, the Critical Appraisals Skills Program Appraisal Checklists, and the Mixed Methods Appraisal Tool were used to assess the quality of the quantitative studies and quantitative components from mixed methods studies [19], qualitative studies and the qualitative components from mixed methods studies [20], and the experiments and mixed methods studies (Appendix A), respectively. Quality of studies can be scored as a percentage depending on how many set criteria are met by the study being assessed (Appendix A). The review protocol is available on PROSPERO (ID: CRD42021244809).

### 2.4. Data Synthesis and Analysis

Data organization was carried out using Microsoft Excel. Statistically significant values were reported to present the direction and range of effect of each relevant factor (Appendix A).

## 3. Results

### 3.1. Search Results and Study Characteristics

Table 1 summarizes the characteristics of the included studies. We identified 30 studies from eight countries. More than 70% (*n* = 22) of the studies were conducted in China (*n* = 18) and USA (*n* = 4). Data collection in half (*n* = 14) of the included studies was conducted between 2010 and 2014. Most studies (*n* = 24) employed cross-sectional designs. There were 29 quantitative (including two experiments) and 1 qualitative study. Twelve studies covered both rural and urban settings. Seventeen focused on actual vaccine uptake, seven on caregivers’ intention for childhood influenza vaccination and six on both. Identified non-biomedical factors of childhood influenza vaccination were categorized and analyzed according to the factors listed in Appendix A and their definitions, including knowledge and attitudes toward influenza vaccine, perceptions of influenza and influenza vaccines, cue to action, emotion, individual characteristics, and contextual factors.

Study quality assessment: Study quality ranged from 19% to 90% with an average score of 62% across all 30 studies. The majority of studies did not involve members of the public in the study design (25), discuss potential response biases (24), and lacked evidence of data dredging (25). In order to provide an overview of the entire literature, no studies were excluded based on their quality.

### 3.2. Childhood Influenza Vaccination and Caregivers’ Intention for Childhood Influenza Vaccination

Substantial variations in coverage of childhood influenza vaccination have been reported across studies. Regionally, childhood influenza vaccination coverage ranged from 6.58% (in the past flu season) in Pakistan [38] to 96.4% (2013–2014 flu season) in Ansan and Jeonju cities, South Korea [44]. Studies in Hong Kong reported higher coverage: 58.9 (in the past flu season)–63.2% (2011–2012 flu season) [40,49]. Other high-coverage areas included Colorado (50.2% in the past flu season), Texas (65%, 2010–2011 flu season), England (52.8%, 2015–2016 flu season), and Guangzhou city, China (47%, 2012–2013 flu season) [36,39,42,48]. Most studies reported high vaccination intention among caregivers to vaccinate their children against influenza. The highest intended acceptance of 92.6% was reported in Ansan and Jeonju cities, South Korea [44], followed by Seoul, South Korea (83.57%) [33].

### 3.3. Influencing Factors of Childhood Influenza Vaccination and Caregivers’ Intention for Childhood Influenza Vaccination

#### 3.3.1. Caregivers’ Knowledge

Measurements for knowledge and related constructs varied. Among the included studies, most of the studies employed single-item questions and summary scores [22,24,29,32,35,38]. There was evidence that caregivers’ knowledge toward influenza vaccines influenced their decisions about whether to vaccinate their children. Caregivers that have higher knowledge about the influenza vaccine were more likely to vaccinate their children (OR = 1.13–2.64) [22,29,35]. Higher knowledge was also associated with a stronger intention to vaccinate their children (OR = 1.74) [32].

Caregivers’ awareness of influenza was reported in four studies, employing single-item questions for measurements [25,29,38,51]. Only one study showed that caregivers’ awareness regarding that “Children should be vaccinated every year” was associated with children’s influenza vaccination in the past flu season (OR = 2.34) [29].

#### 3.3.2. Caregivers’ Attitudes towards Influenza Vaccines

Caregivers’ attitudes toward influenza vaccines, including acceptance of vaccines or willingness to vaccinate children, were widely reported in ten studies [23,25,32,33,34,37,38,42,43,47,51]. Attitudes varied across studies. In China and England, approximately 53.8% and 50.57% of surveyed caregivers deemed the influenza vaccine necessary, respectively [32,42]; meanwhile, about 98% of Thai caregivers reported clear intention to vaccinate their children against influenza [51]. Caregivers with higher acceptance or positive attitudes of influenza vaccine were more likely to vaccinate their children (OR = 1.88–7.46) [23,42,49]. Patterns were similar for associations between caregivers’ attitude regarding the influenza vaccine and their intention to vaccinate their children [32,42].

Caregivers’ self-efficacy in deciding on childhood influenza vaccination was reported in six studies, which all employed preexisting scales for measurement [30,35,39,40,43,49]. Multiple studies revealed most caregivers had a high level of self-efficacy in taking their children for influenza vaccination. The percentage of parents who said they were able to vaccinate children if they desire to do so no matter how difficult ranged from 79% to 96.3%. A high level of self-efficacy was positively associated with caregivers’ likelihood to vaccinate their children (OR =2.96) [39] or intention to vaccinate (OR = 1.25) [30].

#### 3.3.3. Caregivers’ Perceptions of Vaccines

Research frequently explored caregivers’ perceptions of vaccines, which mainly covered four constructs: susceptibility, severity, barriers, and benefits. Six studies employed single-item questions to investigate the above part of constructs, such as perceived benefits and barriers [25,27,29,34,36,38], and 12 studies employed self-made or preexisting scales by summary scores of corresponding answers or self-reported scales [30,31,32,33,35,37,39,40,42,49,50].

Among the included studies, the proportion of parents who perceived a high susceptibility to influenza varied between 10.4% and 83% [34,42]. Meanwhile, 12–82.9% of caregivers believed that influenza was a serious disease [37,50]. Caregivers perceiving more susceptibility and severity to influenza were more likely to vaccinate their children. An increased likelihood of childhood influenza vaccination was observed amongst caregivers who perceived high infection risk (OR = 4.46) and high disease burden (OR = 1.66) [42]. Caregivers’ opinion on the susceptibility (OR = 1.44–3.2) and severity (B = 1.4) of the disease that the influenza vaccine prevents also influenced their intention to vaccinate their children against influenza [32,35,42]. Likewise, believing in vaccine efficacy was often positively associated with caregivers’ likelihood to vaccinate their children (OR = 1.5–4.56) [29,40,42,49] or intention to vaccinate (OR = 1.22–8.85) [30,31,32,35,42,49].

Caregivers’ perceptions of the barriers to vaccination also influenced their decision-making about childhood influenza vaccination. One reason for not vaccinating was concerns about vaccine safety and side effects, as reported by 19.5–61.1% and 19.9–89.8% of caregivers, respectively [25,29,34,35,36,37,38,39,40,42,49,50,51]. Caregivers expressing more concern about safety (OR = 0.16–0.59) or side effects (OR = 0.17–0.26) were less likely to vaccinate their children [29,35,36,42]. Negative association also existed between caregivers’ concern about safety (OR = 0.74, B = −1.78, β = 0.35) or side effects (B = −2.02–−0.53) [32,42,43]. In addition, the poor access to vaccination services, including the cost of the vaccine (OR = 0.84), unmotivated childhood influenza vaccination [23].

Five studies examined the impact of caregivers’ emotions on decisions about childhood influenza vaccination [35,39,40,42,43]. The worry and fear of getting sick supported positive vaccination decisions (OR = 2.31) [39].

#### 3.3.4. Cues to Action and Social Norming

Cues to action and social norming both contain factors related to perceived social pressure from relevant groups or individuals such as family members, friends, and healthcare workers to perform or not perform the behavior. In addition, an individual’s health status or the presence of related symptoms were also factors which could mediate an individual’s perception or even final decision-making about influenza vaccination. Among the included studies, research frequently explored communication about vaccines between caregivers and healthcare workers, family members, and other potential information sources, as well as caregivers’ perception of others’ vaccination behavior (regarding behavior exhibited by others as sensible), and self-rated health status of themselves or children. Studies in Singapore, USA, England, and Thailand all indicated that having had a healthcare worker recommend vaccination (OR = 2.8–8.2, PR = 1.47–2.47) [34,36,42,51] could increase childhood influenza vaccination. Increased intention to get children vaccinated was also observed among caregivers whose health care professionals had recommended vaccination (OR = 1.11) or social influence of family or others (OR = 11.23–21.66) [42,49].

#### 3.3.5. Caregivers Characteristics

Caregivers’ decisions regarding childhood influenza vaccination were sometimes influenced by the caregivers’ or children’s characteristics. Family members’ influenza vaccination history was a frequently studied variable.

Caregivers’ (OR = 5.81–9.1) [35,36,40] or children’s (OR = 3.2–15.54) [42,51] previous influenza vaccination experiences were associated with greater childhood influenza vaccination or vaccination intention (OR = 1.79–4.99) in the current flu season [27,30,32,39,42,50]. In addition, caregivers working in enterprises (OR = 1.86–3.15) [22,29], or hospitals (OR = 2.36) [29], having a male child (OR = 1.45–1.58) [23,24,29], and a higher education degree (OR = 3.9) [22] were more likely to vaccinate their children against influenza.

There was less consensus on the effects of other caregivers’ demographics on childhood influenza vaccination, including location (rural/urban), household registration status, and income. One study indicated positive vaccination behaviors among caregivers living urban areas (OR = 4.89) [21], while another study found caregivers in rural areas were more likely to vaccinate their children (OR = 1.82) [24].

#### 3.3.6. Contextual Factors

Only one study conducted in Jiangsu province, China, investigated the impact of contextual factors, including vaccination service delivery and the number of vaccinators per capita in the local population on childhood influenza vaccination. The study showed that a higher frequency of vaccination services (OR = 1.08) and a greater number of vaccinators (OR = 1.2) had positive associations with local childhood influenza vaccination, while a negative association existed between the availability of vaccination services on weekends and childhood influenza vaccination (OR = 0.88) [21].

## 4. Discussion

This systematic review summarizes the literature on factors influencing childhood influenza vaccination among a variety of populations from a diverse set of geographical and cultural contexts. The findings reveal a wide range of childhood influenza vaccination coverage (4.04–96.4%) and caregivers’ intention (48–85.1%) to vaccinate their children. The results indicate that higher knowledge on influenza vaccine, positive vaccine attitudes, a high level of self-efficacy, perceived high susceptibility and severity of influenza, believing in the efficacy of influenza vaccines, the worry of getting sick, healthcare workers’ recommendations, and previous influenza vaccination experiences may increase the uptake of childhood influenza vaccination. In addition, the main barriers that contributed to caregivers’ vaccine hesitancy in the reviewed studies were the fear of the safety and side effects of the vaccine, as well as poor access to vaccination service.

In general, childhood influenza vaccine coverage rates in high-income countries and regions are higher [34,36,37,40,42,43,44,48,49]. These high vaccination rates may be due to a better vaccination infrastructure and free flu vaccinations; however, it is worth noting that despite the availability of free flu vaccines, vaccination rates in high-income countries and regions are still far from ideal [52]. According to the Vaccine Hesitancy Determinants Matrix, factors that influence vaccination uptake are complex [17]; furthermore, results from a single study depend on when and where the survey was conducted, and the surveyed population. All these factors add to the difficulty of comparing flu vaccination uptake across regions. Further investigation is required to determine the relative importance of these factors.

Our findings are well aligned with theoretical models of health behavior, including the Knowledge, Attitude, and Practices (KAP) model, which posits that relevant knowledge and positive attitudes could lead to positive behavior change [53]. We suggest that health education, and providing adequate, clear, and accessible information to caregivers about influenza infection and vaccines could increase caregivers’ understanding of vaccines. Meanwhile, it is worth noting that despite the high volume of studies, tools used to investigate knowledge levels varied. The use of standardized tools for collecting information about respondents’ knowledge on influenza viruses and vaccines, similar to ones available for other vaccines [54], could facilitate more consistent data collection and enable researchers to more accurately compare the knowledge level and perceptions among people in different countries and regions.

We found that respondents’ perceived susceptibility and severity of influenza and vaccine effectiveness were associated with childhood influenza vaccination and vaccination intention. This finding is consistent with previous studies [55]. Additionally, our systematic review identified concerns about vaccine safety and side effects as the main barriers determining caregivers’ willingness to vaccinate. Communication efforts, such as health communication techniques on a variety of media platforms, are needed to leverage the positive themes that emerged as encouraging high vaccine uptake, including the rigorous safety process in vaccine development and approval by the drug administration authority entities.

Our results showed a strong consensus on the impact of healthcare workers’ (HCWs) recommendations on patient uptake. The importance of healthcare workers as trusted sources of information aligns with previous research [56,57]. Studies have shown wide variations in vaccine recommendation behavior among HCWs among countries and regions, with a low frequency of recommendation practice in China [58,59], and a high frequency of that in the US and European countries [60,61,62,63]. Factors at different levels, including knowledge and confidence in vaccines [62,63], workload, communication skills, financial incentives for recommending vaccines, and whether vaccines are free or not, all influence the recommendation behavior of HCWs [64]. Further research into the factors that influence HCWs’ vaccination recommendation practice is needed to optimize the communication between HCWs and the public.

Among caregivers’ characteristics, our study has shown that influenza vaccination history is a strong predictor of vaccine acceptance, which is consistent with previous studies [65,66,67]. As for contextual factors, we found few studies investigating this factor. As the Vaccine Hesitancy Matrix shows, vaccination behavior is viewed as being affected by multiple levels of familial, social, and cultural influences [68]. The procurement and supply of vaccines, the promotion of vaccines by the health sector and HCWs all could influence individual vaccine decisions. In China, the National Immunization Advisory Committee (NIAC), which has a duty to advise national authorities with evidence-based recommendations on immunization policy and program [69,70,71], considered the influenza vaccine to be beneficial to children in preliminary evaluation. The final decision on whether to include the influenza vaccine in the EPI system depends on the further evaluation of the vaccine using robust frameworks, which requires a substantial amount of resource-intensive scientific work [72]. Future research is needed to focus on changes in influenza vaccination policies and its impact on caregivers’ vaccination decisions.

More importantly, the COVID-19 pandemic, which had an unrivalled impact on global healthcare and social systems, may also affect the public perception of influenza vaccines and even vaccination behavior in different ways [73]. A study showed caregivers indicated that the adoption of nonpharmacologic interventions (NPIs) during COVID-19 reduced the risk of influenza infection for children [74]. In addition, an infodemiology study found that the significantly polarizing nature of public opinions toward COVID-19 vaccination may have adversely affected influenza vaccination sentiments [75]. Thus, further research is required to understand how public perceptions of influenza changed during COVID-19 and to develop more effective and comprehensive strategies to promote vaccination.

The strength of this study is its analysis of research across a range of countries and regions. While caregivers’ knowledge, attitude, and perceptions of vaccines and diseases varied between countries and regions, our review indicated their consistent association with childhood influenza vaccination. Our study has several limitations. First, although this study included 12 countries and regions, it may not be representative of global childhood influenza vaccination. There is limited data from low-income countries, which is a barrier to generalizing our findings. Second, most studies utilized self-reported questionnaires as their main data collection method. Inherently, this approach is most suitable at the given time as such studies are measuring the subjective perception of individuals. However, it is important to note that self-reported surveys have a few limitations, including social desirability bias and recall bias [76,77]. Moreover, caregivers’ perception or attitudes to vaccination are influenced by complex factors, including vaccination price and vaccine safety incidents. Finally, the different measures and outcome variables and inconsistent reporting limited the ability to conduct direct statistical comparisons or draw generalizable conclusions on every predictor. In addition, the amount of research that proceeds publication is limited; studies reporting significant findings are more likely to be published, which may have introduced bias to our review.

## 5. Conclusions

By expanding our knowledge about specific vaccine perceptions and behavior associations and factors hindering or contributing to childhood influenza vaccination, this paper may guide the future study in China and other countries with similar contexts and guide the development of interventions to increase childhood influenza vaccination. The majority of included studies adopted quantitative methods. Only one study utilized qualitative methods. Therefore, more qualitative studies are needed to provide insights into the formation of caregivers’ attitudes and perceptions, allowing deeper understanding beyond predetermined quantitative tools. Our study presents the positive association between childhood influenza vaccination and caregivers’ knowledge on influenza vaccine, positive vaccine attitudes, perceived high susceptibility and severity of influenza, and higher confidence in the influenza vaccine. Hence health education is necessary to inform caregivers and increase their understanding of vaccines. Our study indicates that recommendations from HCWs may increase childhood influenza vaccination. To better motivate HCWs to recommend influenza vaccines and improve the communication between HCWs and the public around vaccination, further research should investigate the influencing factors of HCWs’ recommendation behavior in different countries or regions. In addition, studies on contextual factors, including the procurement and supply of vaccines, and the promotion of vaccines by the health sector are needed to provide insights into how contextual factors can drive public vaccination behaviors.

## Figures and Tables

**Figure 1 vaccines-12-00233-f001:**
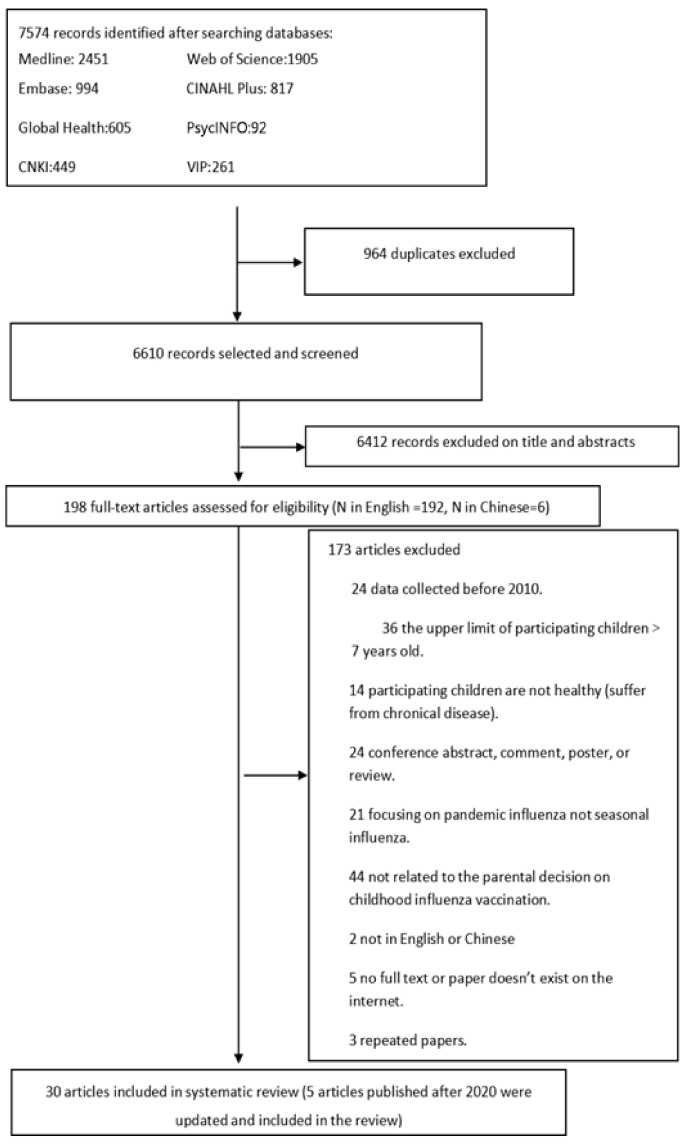
Flowchart of study identification and selection.

**Table 1 vaccines-12-00233-t001:** Summary of characteristics of included studies that investigated factors influencing childhood influenza vaccination.

Characteristic	Number of Studies	Studies
Total	30	
Language		
Chinese	8	[21,22,23,24,25,26,27,28]
English	22	[29,30,31,32,33,34,35,36,37,38,39,40,41,42,43,44,45,46,47,48,49,50,51]
Year of data collection		
2010–2014	14	[25,26,30,33,35,36,37,39,40,43,44,47,49,50]
2015 and later	15	[21,22,23,24,29,31,32,34,42,45,46,48,51]
NR	1	[38]
Study design		
Quantitative study		
Longitudinal	3	[43,48,49]
Cross-sectional	24	[21,22,23,24,25,29,30,31,32,34,35,36,37,38,39,40,41,42,44,47,50,51]
Experiment	2	[33,46]
Qualitative study	1	[45]
Study region		
Asia		
China		
Mainland China	12	[21,22,23,24,25,26,27,28,29,31,32,39]
Hong Kong	5	[35,40,43,49,50]
Taiwan	2	[30,32]
South Korea	2	[33,44]
Singapore	1	[34]
Pakistan	1	[38]
Thailand	1	[51]
Europe		
England	1	[42]
North America		
USA	4	[36,37,47,48]
Oceania		
Australia	2	[45,46]
Urbanicity		
Urban	12	[23,26,28,34,35,40,44,45,46,47,49,50]
Rural	0	
Both (urban and rural)	12	[21,22,24,25,29,30,31,32,42,43,48,51]
Unknown	6	[27,33,36,37,38,39]
Outcome variable		
Vaccination behavior	17	[21,22,23,24,34,35,36,37,38,40,46,48,51]
Vaccination intention	7	[27,30,31,32,33,47,50]
Both	6	[29,39,42,43,44,49]

## Data Availability

The corresponding author had full access to all the data in the study and had final responsibility for the decision to submit for publication. The data presented in this study are openly available.

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
