# Peer review of "Factors Influencing Childhood Influenza Vaccination: A Systematic Review"

_vaccines, 2024, doi:10.3390/vaccines12030233_

Round 1
Reviewer 1 Report
Comments and Suggestions for Authors
The article is certainly relevant, timely and useful. Unfortunately, improving vaccine adherence remains important, which means doctors need to do a better job of communicating with patients. What I didn’t really like about the manuscript. In the division of the studied articles into “2010-2014” and “after 2015”, the COVID-19 period (2019-2022) should be added. During this time, adherence decreased further and it was appropriate to also reflect this in the manuscript. After adding this section, the design of this study will be more properly planned.
Author Response
Response: Many thanks for your comments. As you commented, the COVID-19 pandemic may affect public perception and attitudes toward influenza and influenza vaccines. The adoption of nonpharmacologic interventions (NPIs), such as mandated face coverings in public, have influenced the incidence of influenza. Study showed decreased influenza incidence in 2020 (January through May) after adoption of NPIs as compared with prior seasons.
In this review, five included studies (1. A longitudinal study using parental cognitions based on the theory of planned behavior to predict childhood influenza vaccination; 2. Associated Factors of Behavioral Intention Regarding Childhood Influenza Vaccination Among Parents of Ever-Vaccinated and Never-Vaccinated 24- to 59-Month-Old Children in Hong Kong; 3. Knowledge, attitude/perception, and practice related to seasonal influenza vaccination among caregivers of young Thai children: A cross-sectional study; 4. An Investigation on Influenza Awareness and Influenza vaccination willingness of parents of Preschool Children; 5. Influenza Vaccination Among Preschool Children in Nanshan District of Shenzhen.) were published after 2020, but all the data were collected before 2020, meaning that in these studies, the occurrence of COVID-19 did not affect the decision-making of children caregivers. Therefore, it makes little sense to include "the COVID-19 period" in the division of the studies.
Furthermore, I applied the latest relevant literature push function through the London school’s literature search platform, which can push the latest relevant literature in those databases to my email. Therefore, after completing the first round of literature search and screening, New relevant literatures can still be included in the review. I also explained this in the article. However, the number of included literatures published after 2020 is very small. I think the reason may be that most researchers have focused on COVID-19 and COVID-19 vaccines after the outbreak of COVID-19.
Reviewer 2 Report
Comments and Suggestions for Authors
This manuscript deals with influenza vaccination problems from children from several World regions, mostly China and USA with a adequate systematic revision analysis. The main concern of the article is the absence of comparison of the countries policies in the study areas as several unequal regions were studied. Thus the main conclusions of the article are the obvious answers for the problem. Caregivers perceptions and valoring of vaccination are the most impacting determinants. The cost of vaccination was not discussed in the text and we believe that the cost of vaccination could impact in the decision of vaccinate. Please insert some comments of this fact in the manuscript.
Comments on the Quality of English LanguageThere are some mistakes in English as in line 42-43. Please revise
Author Response
Response: Many thanks for your comments. The main purpose of this systematic review was to identify the influencing factors involved in the existing studies on childhood influenza vaccination, summarize the gaps in the existing studies and further inform my PhD study. I conducted a brief analysis of the childhood influenza vaccination uptake rate between different countries and regions. It shows that childhood influenza vaccination rates in developed countries and regions (e.g., South Korea, Hong Kong) are generally higher than developing ones (e.g., mainland China). It can be expected that the countries policies will influence the access to vaccination service and, ultimately, public’s decision-making about vaccination. However, it is difficult to make an in-depth comparison of the countries policies in the study areas. The results of the studies included did not contain enough statistically significant data on local policies to support further comparisons. In the second paragraph of my discussion, I briefly described the role played by country policies and the price of vaccination services. It reads: "These high vaccination rates may be due to reflect a better vaccination infrastructure and free flu vaccinations in some places, however, it is worth noting that despite the availability of free flu vaccines, vaccination rates in high-income countries and regions are still far from ideal.”
I also made some revision in the discussion section to emphasize the significant role of vaccine policy. Meanwhile, including influenza vaccine into national immunization program requires sufficient evidence.
Line 338-354: Furthermore, the introduction of the influenza vaccine into the list of National immunization program (NIP) can also effectively reduce health expenditure. Decisions for vaccine introduction must be based on scientific evidence. Inclusion of vaccines in the NIP system requires sufficient evidence in five areas, including: features of the disease (epidemiological characteristics and disease burden); features of the vaccine (vaccine characteristics performance, and cost-effectiveness); ability of the vaccination to be implemented in the NIP system (Availability of vaccine supply, financial issues, and human resource and infrastructure); international experience with the vaccine (WHO recommends, Experience of other countries); and potential societal impact of the vaccine(acceptability, and ethical consideration)[68]. In China, for example, the National Immunization Advisory Committee (NIAC), established in 2017, has a duty to advise national authorities with evidence-based recommendations on immunization policy and program [69, 70]. Preliminary evaluation showed that experts in NIAC consider influenza vaccine to be beneficial to children. The final decision on whether to include influenza vaccine in the EPI system depends on the further evaluation of the vaccine using robust frameworks, which requires a substantial amount of resource-intensive scientific work[71].
It should be emphasized that the inability to make comparison of the countries policies in the study areas does not affect my use of the results of this systematic review to inform the further work of my PhD study.
Reviewer 3 Report
Comments and Suggestions for Authors
Reading this manuscript, I noted a number of concerns and a general paucity of insights with the analysis.
Specific comments:
1. As per the journal's guidelines to authors, the abstract should be around 200 words and be structured but without subheadings.
2. In the introduction section, the reported influenza vaccination coverage for the different countries need to be updated and should include a range of rates covering the whole study period. Accordingly, the references need to be updated too.
3. The introduction does not sufficiently explain the specific focus on childhood influenza vaccination. Elderly too are susceptible to severe complications from influenza, including pneumonia, bronchitis, etc.
4. The search results are more than 3 years old now. Authors should update the search results to ensure that the findings are timely, relevant and uptodate.
5. The protocol deposited in PROSPERO differs from what is written in the methods section. Details such as "1) parental decision (acceptance, willingness, intention or past behavior) on seasonal influenza vaccine and influencing factors, for example, attitude, perceptions to seasonal influenza vaccine or 2) migrants' decision on seasonal influenza vaccine , Covid-19 vaccine and influencing factors" are nowhere to be found in the current manuscript, neither is the focus on COVID-19 vaccination among children which is part of the research objective according to the protocol given in PROSPERO. Please update the public record to ensure it is accurate and concordant.
6. "PshchInfo" is misspelled in Figure 1. Suggest also using the latest PRISMA flowchart template please.
7. Please reformat Table 1, it is extremely hard to interpret. What exactly is "Both" or "Unknown" referring to??
8. "Hongkong" should be written as "Hong Kong" and "South-Korea" as "South Korea".
9. A more qualitative synthesis is needed in the results section. A diagrammatic representation classifying factors at individual, group, and societal levels might offer better clarity. The current structure doesn't effectively encapsulate the literature's findings or provide novel insights.
10. It is unfortunate that the discussion section is still heavily focused on KAP even though the authors stated in the introduction, as a motivation for the current review, "However, previous studies only focused on the factors at the individual level, including knowledge of influenza, awareness of risk, misconceptions regarding vaccine safety and efficacy." Please probe more deeply into the findings. In terms of legislation as well, countries vary in their approach to influenza immunization for children. Some countries include the influenza vaccine in their national immunisation schedules for kids, offering it free of charge. Not all countries have made it a standard part of their national schedules, and the cost, vaccine availability, and governmental perceptions of the vaccine's necessity can be rather variable. Why is it that some countries and not others have included the influenza vaccine in national immunisation schedules for children?
11. Beyond issues on vaccine hesitancy in general, it would also be relevant to mention the rise in vaccination hesitancy in relation to the COVID-19 pandemic. Several studies have found potential negative spill-over effects from the pandemic, with particular coincides with misinformation related to COVID-19 policies and vaccination (citation: pubmed.ncbi.nlm.nih.gov/37376407). These are also relevant to consider.
12. Please mention in the data availability statement whether the data supporting the article can be made publicly available.
Comments on the Quality of English LanguageMinor edits necessary.
Author Response
Specific comments:
- As per the journal's guidelines to authors, the abstract should be around 200 words and be structured but without subheadings.
Response: Many thanks for your comments. I have revised abstract to conform to the journal's guidelines. The revised abstract is shown below:
Childhood influenza vaccination coverage remains low in lower/middle income countries. This systematic review aims to identify influencing factors around childhood influenza vaccination. A systematic literature review was conducted and included empirical studies with original data that investigated factors influencing childhood influenza vaccination. We searched MEDLINE, Web of Science, EMBASE, CINAHL Plus, Global health, and PsycINFO and, two Chinese databases (China Knowledge Resource Integrated Database and Chongqing VIP), using a combination of the key terms ‘childhood’, ‘influenza’, ‘vaccination’, and related syntax for all peer-reviewed publications published before December 2019. Thirty studies were included in the analysis. Childhood influenza vaccination was positively associated with caregivers’ knowledge of influenza vaccine, positive vaccine attitudes, self-efficacy, perceived susceptibility, and severity of influenza, believing in the efficacy of influenza vaccine, the worry of getting sick, healthcare worker’s recommendation and previous influenza vaccination experiences. Barriers included the fear of safety and side effects of the vaccine, as well as poor access to vaccination service. To improve childhood influenza vaccine uptake, health education is necessary to inform caregivers and increase their understanding of vaccines. Future studies are needed to investigate influencing factors around healthcare workers vaccination recommendation behaviors and the impact of contextual factors on public vaccination behaviors.
- In the introduction section, the reported influenza vaccination coverage for the different countries need to be updated and should include a range of rates covering the whole study period. Accordingly, the references need to be updated too.
Response: Many thanks for your comments. There is a lack of data on influenza vaccination rates among children under five years. In China, for example, the data with national representativeness on childhood influenza vaccination were collected in five provinces in 2009-12, and I have updated them into this review.
Line 54-56: Data collected between 2009 and 2012 indicate that influenza vaccination uptake among children <5 years living in five provinces in mainland China was about 26.4%, which is far from satisfactory [15].
- The introduction does not sufficiently explain the specific focus on childhood influenza vaccination. Elderly too are susceptible to severe complications from influenza, including pneumonia, bronchitis, etc.
Response: Many thanks for your comments. According to recommendation of the World Health Organization (WHO), yearly seasonal influenza vaccination is especially important for people at high risk of influenza complications and their carers, including pregnant women, children aged 6 months to 5 years, people over age 65, people with chronic medical conditions and health workers.Although influenza is often regarded as an illness of the elderly population due to the highest influenza-related excess mortality among persons over 65 years of age, ample evidence indicates that the burden of influenza is also substantial in children. Children have the highest rates of infection in the community during epidemics. It causes great disease burden among children below 5 years of age, with an estimated 870,000 hospitalizations and 10,200 deaths per year worldwide. I choose to focus on caregivers’ decision-making on childhood influenza vaccination because vaccinating young children (6 to 60 months) is an important measure, which can provide individual protection and reduce transmission across all age groups thereby decreasing the disease burden across the population. I have made corresponding changes in the manuscript.
Line 41-43: Children have the highest rates of infection in the community during epidemics. It causes great disease burden among children below 5 years of age, with an estimated 870,000 hospitalizations and 10,200 deaths per year worldwide[5].
- The search results are more than 3 years old now. Authors should update the search results to ensure that the findings are timely, relevant and uptodate.
Response: Many thanks for your comments. As mentioned in the manuscript, the deadline for the first round of article search is 2020. However, at the same time, I have applied the latest relevant literature push function through the london school’s literature search platform, which can push the latest relevant literature in those databases to my email. Therefore, after completing the first round of literature search and screening, New relevant literatures can still be included in the review. I also explained this in the article. However, the number of included literatures published after 2020 is very small. I think the reason may be that most researchers have focused on COVID-19 and COVID-19 vaccines after the outbreak of COVID-19. So I can confirm that this review covers the most recent literatures on caregiver coverage of childhood influenza vaccination
- The protocol deposited in PROSPERO differs from what is written in the methods section. Details such as "1) parental decision (acceptance, willingness, intention or past behavior) on seasonal influenza vaccine and influencing factors, for example, attitude, perceptions to seasonal influenza vaccine or 2) migrants' decision on seasonal influenza vaccine , Covid-19 vaccine and influencing factors" are nowhere to be found in the current manuscript, neither is the focus on COVID-19 vaccination among children which is part of the research objective according to the protocol given in PROSPERO. Please update the public record to ensure it is accurate and concordant.
Response: Many thanks for your comments. The main purpose of this systematic review was to identify the influencing factors involved in the existing studies on childhood influenza vaccination, summarize the gaps in the existing studies and further inform my PhD study. However, the COVID-19 outbreak in 2020 makes it difficult to complete data collection. Therefore, I consider changing my doctoral project to focus on migrants' decision-making on seasonal influenza and COVID-19 vaccination. And complete the data collection through online means. At the time of registration on PROSPERO, I combined the two research topic because I have almost finished the data extraction for the review on caregivers' decision-making about childhood flu vaccines. I informed the PROSPERO staff responsible for reviewing registration documents with the reason for combining two topics, and they allowed it. Later, COVID-19 in the mainland was brought under control, and I successfully completed the original data collection plan, so the second topic was not done. I have updated the PROSPERO content as required.
- "PshchInfo" is misspelled in Figure 1. Suggest also using the latest PRISMA flowchart template please.
Response: Many thanks for your comments. I have made corresponding revisions.
- Please reformat Table 1, it is extremely hard to interpret. What exactly is "Both" or "Unknown" referring to??
Response: Many thanks for your comments. I have made corresponding revisions to make this table easy to interpret.
- "Hongkong" should be written as "Hong Kong" and "South-Korea" as "South Korea".
Response: Many thanks for your comments. I have made changes accordingly.
- A more qualitative synthesis is needed in the results section. A diagrammatic representation classifying factors at individual, group, and societal levels might offer better clarity. The current structure doesn't effectively encapsulate the literature's findings or provide novel insights.
Response: Many thanks for your comments. The main purpose of this systematic review was to identify the influencing factors involved in the existing studies on childhood influenza vaccination, summarize the gaps in the existing studies and further inform my PhD study. Therefore, I choose the descriptive analysis method, because the existing analysis methods can meet the purpose of this research. The analysis method and the classification of influencing factors have been used by other study (doi: 10.1136/bmjgh-2020-003599) and proved to be sufficient to present the findings of the literature.
- It is unfortunate that the discussion section is still heavily focused on KAP even though the authors stated in the introduction, as a motivation for the current review, "However, previous studies only focused on the factors at the individual level, including knowledge of influenza, awareness of risk, misconceptions regarding vaccine safety and efficacy." Please probe more deeply into the findings. In terms of legislation as well, countries vary in their approach to influenza immunization for children. Some countries include the influenza vaccine in their national immunisation schedules for kids, offering it free of charge. Not all countries have made it a standard part of their national schedules, and the cost, vaccine availability, and governmental perceptions of the vaccine's necessity can be rather variable. Why is it that some countries and not others have included the influenza vaccine in national immunisation schedules for children?
Response: Many thanks for your comments. Since most existing studies focus on caregivers' perceptions of vaccines and diseases, the results and discussion sections are less about countries policies.
As for the different policies on childhood influenza vaccination in each country, whether to include the influenza vaccine in the national immunization schedules is a lengthy decision-making process, the data in this systematic review is not sufficient for further discussion, and interviews with local decision-makers responsible for health technology assessment are needed. In China, for example, the National Health Commission (NHC) has a duty to make evidence-based decisions regarding further expansion of the EPI and the replacement of current EPI vaccines with new ones. Decisions for vaccine introduction must be based on scientific evidence. According to experts’ opinions, inclusion of vaccines in the EPI system requires sufficient evidence in five areas, including: features of the disease (epidemiological characteristics and disease burden); features of the vaccine (vaccine characteristics performance, and cost-effectiveness); ability of the vaccination to be implemented in the EPI system (Availability of vaccine supply, financial issues, and human resource and infrastructure); international experience with the vaccine (WHO recommends, Experience of other countries); and potential societal impact of the vaccine(acceptability, and ethical consideration). Preliminary evaluation showed that experts in The National Immunization Advisory Committee (NIAC) consider influenza vaccine to be beneficial to children. The final decision on whether to include influenza vaccine in the EPI system depends on the further evaluation of the vaccine using robust frameworks.
It could be seen that the decision to include vaccines in the EPI which requires a substantial amount of resource-intensive scientific work. Therefore, we can directly describe different countries’ vaccine policies and make simple comparisons, but the decision-making paths behind those policies in those countries require a lot of additional data to analyze. In this regard, I add relevant content in the manuscript and point out that future research needs to perfect the evidence needed for the inclusion of vaccines in EPI.
Line 337-353:Furthermore, the introduction of the influenza vaccine into the list of National immunization program (NIP) can also effectively reduce health expenditure. Decisions for vaccine introduction must be based on scientific evidence. Inclusion of vaccines in the NIP system requires sufficient evidence in five areas, including: features of the disease (epidemiological characteristics and disease burden); features of the vaccine (vaccine characteristics performance, and cost-effectiveness); ability of the vaccination to be implemented in the NIP system (Availability of vaccine supply, financial issues, and human resource and infrastructure); international experience with the vaccine (WHO recommends, Experience of other countries); and potential societal impact of the vaccine(acceptability, and ethical consideration)[70]. In China, for example, the National Immunization Advisory Committee (NIAC), established in 2017, has a duty to advise national authorities with evidence-based recommendations on immunization policy and program [71, 72]. Preliminary evaluation showed that experts in NIAC consider influenza vaccine to be beneficial to children. The final decision on whether to include influenza vaccine in the EPI system depends on the further evaluation of the vaccine using robust frameworks, which requires a substantial amount of resource-intensive scientific work[73].
- Beyond issues on vaccine hesitancy in general, it would also be relevant to mention the rise in vaccination hesitancy in relation to the COVID-19 pandemic. Several studies have found potential negative spill-over effects from the pandemic, with particular coincides with misinformation related to COVID-19 policies and vaccination (citation: pubmed.ncbi.nlm.nih.gov/37376407). These are also relevant to consider.
Response: Many thanks for your comments. There is no denying that the outbreak of COVID-19 have an impact on the other vaccines, especially the flu vaccine. Though the COVID-19 virus and influenza are vastly different pathogens, there are important areas of overlap. For example, the majority of COVID-19 patients present with influenza-like illness. Meanwhile, the adoption of nonpharmacologic interventions (NPIs), such as mandated face coverings in public, have influenced the incidence of influenza.
According to the paper you provided, the significantly polarizing nature of public opinions toward COVID-19 vaccination may have adversely affected influenza vaccination sentiments, and further negatively influence influenza vaccination. According to our interviews with medical staff at local vaccination clinics and child caregivers, there is a general consensus that if influenza vaccination protects against influenza, it should also protect against other respiratory diseases, including COVID-19. It can be seen that the COVID-19 outbreak could have different impacts on residents' perceptions of influenza and influenza vaccination, and the specific reasons may be related to different information dissemination policies and COVID-19 prevention measures between countries and regions. Further research is needed. I have made corresponding changes in the discussion section of the manuscript.
Line 354-363:More importantly, the COVID-19 pandemic, which had an unrivalled impact on global healthcare and social systems, may also affect public perception of influenza vaccines and even vaccination behavior in different ways [74]. Study showed caregivers indicated that the adoption of nonpharmacologic interventions (NPIs) during COVID-19 reduced the risk of influenza infection for children[75]. In addition, an infodemiology study found that the significantly polarizing nature of public opinions toward COVID-19 vaccination may have adversely affected influenza vaccination sentiments [76]. Thus, further research is required to understand how public perceptions of influenza changed during COVID-19 and develop more effective and comprehensive strategies to promote vaccination.
- Please mention in the data availability statement whether the data supporting the article can be made publicly available.
Response: Many thanks for your comments. I have made corresponding revision in the data availability statement.
Reviewer 4 Report
Comments and Suggestions for Authors
This is an interesting paper addressing an important subject area. Novelty is explained. A few suggestions:
1. Change 'developing countries' to 'lower/middle income countries.'
2. Give more details of data synthesis in the methods sections. E.g. how were data from the different health models synthesized?
3. Sometimes exact statistics form a single study are quoted but there are a number of references to those statistics.
4. Consider changing terminology such as 'better knowledge' to 'higher knowledge'
Comments on the Quality of English Language
A few grammatical errors
Author Response
- Change 'developing countries' to 'lower/middle income countries.'
Response: Many thanks for your comments. I have made corresponding changes in the manuscript.
Line 11-12: ”Influenza vaccine coverage in children remains low in lower/middle income countries.”
Line 63-66: ”The findings can inform further studies on factors influencing childhood influenza vaccination in lower/middle income countries, like China, and finally contribute to country-level policy decisions and improve childhood influenza vaccination uptake.”
- Give more details of data synthesis in the methods sections. E.g. how were data from the different health models synthesized?
Response: Many thanks for your comments. I conducted descriptive analyses of data from the included studies. The theories used to guide the conduct of researches were different in included studies, including the Health Belief Model, the‘Knowledge, Attitude, and Practices’ model, and the theory of planned behaviour. The theories contained different constructs. Therefore, the inclusion of studies that use different theories can deepen our understanding of the issue of caregivers’ decision-making about childhood influenza vaccination. As for how the data synthesis was conducted, as I said before, the main purpose of this systematic review was to identify the influencing factors involved in the existing studies on childhood influenza vaccination, summarize the gaps in the existing studies and further inform my PhD study. Therefore, I only used descriptive analyses and did not conduct in-depth analyses on the association between influencing factors and childhood influenza vaccination. A standardized form based on behavioral theories including the ‘Knowledge, Attitude, and Practices’ model, the Health Belief Model and the theory of planned behaviour was developed specifically for this review prior to data extraction. Topics in the form include knowledge and awareness of influenza and influenza vaccines, confidence in the importance, safety, and efficacy of influenza vaccines, perceived susceptibility and severity of influenza, benefits and barriers to influenza vaccination, cue to action and social norm, self-efficacy, and emotions. I extracted the results of the variables in the included studies under the corresponding topics, including the decision on vaccination of respondents in different groups under variables, crude association between variables and childhood influenza vaccination in univariate analysis, and adjusted association in multivariate analysis. Furthermore, I also extracted the results of the association between the socio-demographic and socio-economic factors of the respondents and childhood influenza vaccination, including the age and gender of the child, education level of the parents, the relationship between the caregivers and the child, the residential area, Household registration status, the occupation of the parents, the annual household income, and health insurance status. Factors that were examined as the predictors of influenza vaccination uptake among respondents are presented in the results section. The details are as shown in lines 102-116 in the manuscript.
- Sometimes exact statistics form a single study are quoted but there are a number of references to those statistics.
Response: Many thanks for your comments. I have reviewed the article to make sure that there are no such mistakes already
- Consider changing terminology such as 'better knowledge' to 'higher knowledge'
Response: Many thanks for your comments. I have made corresponding changes in the manuscript. In addition, I have asked a special person to revise the language of this article to make the wording more reasonable.
Line 173-178: ”There was evidence that caregivers’ knowledge toward influenza vaccines influenced their decisions about whether to vaccinate their children. Caregivers having higher knowledge about influenza vaccine were more likely to vaccinate their children (OR = 1.13–2.64) [19,26,32]. Higher knowledge was also associated with stronger intention to vaccinate their children (OR = 1.74) [29].”
Line 276-280: ”The results indicate that higher knowledge on influenza vaccine, positive vaccine attitudes, high level of self-efficacy, perceived high susceptibility and severity of influenza, believing in the efficacy of influenza vaccines, the worry of getting sick, healthcare worker’s recommendation and previous influenza vaccination experiences may increase the uptake of childhood influenza vaccination.”
Round 2
Reviewer 3 Report
Comments and Suggestions for Authors
There are still several issues and problems with the current iteration of the manuscript.
1. Although the authors explained in the introduction and their replies that the "Previous studies have primarily focused on individual level factors, including knowledge of influenza, awareness of risk, misconceptions regarding vaccine safety and efficacy", all the findings synthesized in the review are still limited to the individual level (both intra- and inter-personal level, e.g. cue to action and social norms). Is this an accurate representation of the literature? The main conclusions of the article are the obvious answers for the problem, i.e. caregivers perceptions and knowledge of vaccination are the key determinants for childhood vaccination. This is pretty much the same as your starting point that individual level factors are the answer. So many more factors at the institutional level, community level and policy level influence the decision to vaccinate and these were totally absent from the manuscript. I would suggest a rewriting and reframing of the findings, if it is too difficult to synthesize and make an in-depth comparison of the countries policies in the study areas then I would suggest focusing on just one country. Since majority of the studies found in this review came from China, the authors may consider presenting this as a "Factors influencing childhood influenza vaccination in China: a systematic review" instead.
2. The manuscript still needs a proofreading for language. Some sentences exhibit grammatical inconsistencies, such as incorrect verb tenses and awkward phrasing. An example is in the Introduction: "Children have the highest rates of infection in the community during epidemics. It causes great disease burden..."
3. "... studies related to childhood influenza vaccination published later was pushed through those databases and included in the review for analysis after being confirmed to meet the criteria" - what exactly does this mean? So what is the exact study period for this review? Please state whether it is up till November, 2023 for example.
4. Although the authors claimed that majority of the studies available were quantitative in design, with only one qualitative study found, could it be the result of the inclusion criteria for the review? It is stated that "Studies that focused only on knowledge, attitudes, and beliefs regarding childhood influenza vaccination, but did not refer to actual vaccine uptake were excluded." Does this not inadvertently exclude qualitative studies that did not measure actual vaccine uptake but simply interrogated contextual factors?
Comments on the Quality of English LanguageModerate edits required.
Author Response
1. Although the authors explained in the introduction and their replies that the "Previous studies have primarily focused on individual level factors, including knowledge of influenza, awareness of risk, misconceptions regarding vaccine safety and efficacy", all the findings synthesized in the review are still limited to the individual level (both intra- and inter-personal level, e.g. cue to action and social norms). Is this an accurate representation of the literature? The main conclusions of the article are the obvious answers for the problem, i.e. caregivers perceptions and knowledge of vaccination are the key determinants for childhood vaccination. This is pretty much the same as your starting point that individual level factors are the answer. So many more factors at the institutional level, community level and policy level influence the decision to vaccinate and these were totally absent from the manuscript. I would suggest a rewriting and reframing of the findings, if it is too difficult to synthesize and make an in-depth comparison of the countries policies in the study areas then I would suggest focusing on just one country. Since majority of the studies found in this review came from China, the authors may consider presenting this as a "Factors influencing childhood influenza vaccination in China: a systematic review" instead.
Response: Many thanks for your comments. First, it is not that non-individual factors are not mentioned in this review, but there are only few studies involving these factors, and the results show that only one study showed higher frequency of vaccination services and greater number of vaccinators had positive associations with local childhood influenza vaccination.
As for the differences in childhood influenza vaccination under different countries' vaccination policies, first, the comparison is very difficult, because there are many factors affecting childhood vaccination, including caregivers' perception of the disease and vaccine, different information sources (etc., medical staff, social media), and the access of vaccination services (etc., the price of vaccination, the convenience of transportation, the inventory of vaccines), and these factors could interact with each other. For example, studies have shown that caregivers have a high confidence in vaccines included in the NIP, because they trust government departments and HCWs. In other words, vaccination policies have an impact on caregivers' confidence in vaccines. Therefore, exploring the impact of policy on childhood influenza vaccines is complex. In addition, none of the studies included in the review were representative of the countries in which the data were collected, also making analysis difficult.
As for the suggestion to include only studies from China, I need to explain that this review is part of the doctoral project to confirm the progress of existing studies in caregivers' decision making on childhood influenza vaccination. At the beginning, I also intended to include only the studies in China, but with the progress of the project, it was found that the number of included studies was small. In order to show the existing research progress as much as possible, I chose to include the studies of various countries.
It is also important to note that as well as identifying caregivers' perceptions of disease and vaccines as important influencing factors on childhood influenza vaccination, this review also identified professional information sources, including HCWs. I have therefore conducted a survey to identify factors influencing HCWs' vaccine recommendation behaviors, and this research has been published. I will also analyze the impact of vaccination policies on vaccination behaviors as reviewer suggested in future studies.
2.The manuscript still needs a proofreading for language. Some sentences exhibit grammatical inconsistencies, such as incorrect verb tenses and awkward phrasing. An example is in the Introduction: "Children have the highest rates of infection in the community during epidemics. It causes great disease burden..."
Response: Many thanks for your comments. I have made changes in the manuscript, including:
Line 24 : health education is necessary to address caregivers’ lack of confidence on vaccine safety.
Line 37 : One of the particularly vulnerable groups is young children, as they have the highest rates of infection in the community [4]. Influenza viruses causes great disease burden among children below 5 years of age, with an estimated 870,000 hospitalizations and 10,200 deaths per year worldwide [5].
Line 54: The WHO 's Strategic Advisory Group of Immunization (SAGE) proposed in the Vaccine Hesitancy Determinants Matrix that individual/social influences, contextual influences and vaccine and vaccination-specific issues all play a role [17].
Line 318: These high vaccination rates may be due to a better vaccination infrastructure and free flu vaccinations.
Line 323: furthermore, results from a single study depend on when and where the survey was conducted, and the surveyed population.
Line 338: We found that respondents’ perceived susceptibility and severity of influenza, and vaccines effectiveness was associated with childhood influenza vaccination and vaccination intention.
Line 349: Studies have shown wide variations in vaccine recommendation behavior among HCWs among countries and region, with low frequency of recommendation practice in China [59, 60], and high frequency of that in US and European countries [61-64].
3. "... studies related to childhood influenza vaccination published later was pushed through those databases and included in the review for analysis after being confirmed to meet the criteria" - what exactly does this mean? So what is the exact study period for this review? Please state whether it is up till November, 2023 for example.
Response: Many thanks for your comments. In the first round of literature search, I logged in each database through the Intranet of London School of Hygiene and Tropical Medicine. Keywords for literature search were reserved in each database, and the corresponding new studies were updated on a daily basis. The updated article list is sent to my email address. I will review the title of the article and abstract to determine whether it is necessary to further read the full text. Through this process, I have included 5 new literatures after 2020. The literature was updated and reviewed until the submission of this review, which is in September 2023.
4.Although the authors claimed that majority of the studies available were quantitative in design, with only one qualitative study found, could it be the result of the inclusion criteria for the review? It is stated that "Studies that focused only on knowledge, attitudes, and beliefs regarding childhood influenza vaccination, but did not refer to actual vaccine uptake were excluded." Does this not inadvertently exclude qualitative studies that did not measure actual vaccine uptake but simply interrogated contextual factors?
Response: Many thanks for your comments. For qualitative studies we found in the first round of literature search, we would read the literature to determine whether the researcher started from caregivers' decisions about children's influenza vaccination and further explore their reasons for choosing to vaccinate children against influenza or not. For example, a qualitative study would be excluded if researchers simply interviewed caregivers and asked them about their views on flu and the flu vaccine. Whereas, a study would have been included if they had ascertained the child's influenza vaccination status (Yes or No) beforehand and asked caregivers why. With regard to the fact that there was only one qualitative article selected, I think the reason for this is mainly due to the small number of studies that used qualitative methods compared to those that used quantitative methods.

Round 3
Reviewer 3 Report
Comments and Suggestions for Authors
"For example, a qualitative study would be excluded if researchers simply interviewed caregivers and asked them about their views on flu and the flu vaccine. Whereas, a study would have been included if they had ascertained the child's influenza vaccination status (Yes or No) beforehand and asked caregivers why. With regard to the fact that there was only one qualitative article selected, I think the reason for this is mainly due to the small number of studies that used qualitative methods compared to those that used quantitative methods."
The authors seemed to have either misunderstood my point or not be well-versed with qualitative research. In qualitative research, the emphasis is typically on understanding the depth and complexity of human behavior and decision-making processes. Such studies might delve into the reasons behind caregivers' attitudes towards vaccination, or explore broader social and cultural factors, without necessarily documenting whether these attitudes translated into actual vaccination of the children. Therefore, the authors' criteria for including qualitative studies might have led to a narrower focus, potentially overlooking rich, contextual data that could inform a more nuanced understanding of the factors influencing childhood influenza vaccination.
Comments on the Quality of English LanguageMinor edits only.
Author Response
Thank you for your comments. I apologize for my unclear explanation of your comments in the last round. I totally agree with your explanation of qualitative research. But when I say "a qualitative study would be excluded if researchers simply interviewed caregivers and asked them about their views on flu and the flu vaccine ', I don't mean to explore whether these attitudes would translate into actual vaccination of the children. Rather, we want to include studies that obtain a range of perspectives and achieve variation in terms of childhood influenza vaccination status. According to the comments of examiner, my colleague and I reviewed the qualitative studies included in the first round again, and we can confirm that all relevant qualitative studies have been included. The reason for the small number of included studies is that the number of qualitative studies is very small.